# Radiomics of Liver Metastases: A Systematic Review

**DOI:** 10.3390/cancers12102881

**Published:** 2020-10-07

**Authors:** Francesco Fiz, Luca Viganò, Nicolò Gennaro, Guido Costa, Ludovico La Bella, Alexandra Boichuk, Lara Cavinato, Martina Sollini, Letterio S. Politi, Arturo Chiti, Guido Torzilli

**Affiliations:** 1Department of Nuclear Medicine, Humanitas Clinical and Research Center-IRCCS, 20089 Rozzano-Milan, Italy; francesco.fiz.nm@gmail.com (F.F.); martina.sollini@hunimed.eu (M.S.); arturo.chiti@hunimed.eu (A.C.); 2Division of Hepatobiliary and General Surgery, Department of Surgery, Humanitas Clinical and Research Center–IRCCS, 20089 Rozzano-Milan, Italy; guidocmcosta@gmail.com (G.C.); guido.torzilli@hunimed.eu (G.T.); 3Department of Biomedical Sciences, Humanitas University, 20089 Pieve Emanuele-Milan, Italy; drgennaro.med@gmail.com (N.G.); ludovico.labella@st.hunimed.eu (L.L.B.); alexandra.boichuk@st.hunimed.eu (A.B.); letterio.politi@childrens.harvard.edu (L.S.P.); 4Department of Radiology, Humanitas Clinical and Research Center-IRCCS, 20089 Rozzano-Milan, Italy; 5MOX Laboratory, Department of Mathematics, Politecnico di Milano, 20133 Milan, Italy; lara.cavinato@polimi.it

**Keywords:** radiomics, texture analysis, computer-assisted diagnosis, liver metastases, gray level matrices, response to chemotherapy, overall and recurrence-free survival

## Abstract

**Simple Summary:**

Patients with liver metastases can be scheduled for different therapies (e.g., chemotherapy, surgery, radiotherapy, and ablation). The choice of the most appropriate treatment should rely on adequate understanding of tumor biology and prediction of survival, but reliable biomarkers are lacking. Radiomics is an innovative approach to medical imaging: it identifies invisible-to-the-human-eye radiological patterns that can predict tumor aggressiveness and patients outcome. We reviewed the available literature to elucidate the role of radiomics in patients with liver metastases. Thirty-two papers were analyzed, mostly (56%) concerning metastases from colorectal cancer. Even if available studies are still preliminary, radiomics provided effective prediction of response to chemotherapy and of survival, allowing more accurate and earlier prediction than standard predictors. Entropy and homogeneity were the radiomic features with the strongest clinical impact. In the next few years, radiomics is expected to give a consistent contribution to the precision medicine approach to patients with liver metastases.

**Abstract:**

Multidisciplinary management of patients with liver metastases (LM) requires a precision medicine approach, based on adequate profiling of tumor biology and robust biomarkers. Radiomics, defined as the high-throughput identification, analysis, and translational applications of radiological textural features, could fulfill this need. The present review aims to elucidate the contribution of radiomic analyses to the management of patients with LM. We performed a systematic review of the literature through the most relevant databases and web sources. English language original articles published before June 2020 and concerning radiomics of LM extracted from CT, MRI, or PET-CT were considered. Thirty-two papers were identified. Baseline higher entropy and lower homogeneity of LM were associated with better survival and higher chemotherapy response rates. A decrease in entropy and an increase in homogeneity after chemotherapy correlated with radiological tumor response. Entropy and homogeneity were also highly predictive of tumor regression grade. In comparison with RECIST criteria, radiomic features provided an earlier prediction of response to chemotherapy. Lastly, texture analyses could differentiate LM from other liver tumors. The commonest limitations of studies were small sample size, retrospective design, lack of validation datasets, and unavailability of univocal cut-off values of radiomic features. In conclusion, radiomics can potentially contribute to the precision medicine approach to patients with LM, but interdisciplinarity, standardization, and adequate software tools are needed to translate the anticipated potentialities into clinical practice.

## 1. Introduction

The liver is a frequent target for distant metastases from several tumors. Liver metastases (LM) are associated with poor prognosis and may occur early in gastrointestinal malignancies because of hematogenous spread through the portal venous system [1,2,3,4,5]. Selected patients with LM, mainly those with liver-only metastases, can be considered for aggressive systemic and loco-regional treatments to prolong survival expectancy and optimize quality of life. Several studies have focused on LM from colorectal cancer, for which significant progress has been achieved. Effective chemotherapy may lead to a relevant improvement in survival, exceeding 30 months in the most favorable reports [6,7,8]. Liver resection in selected patients obtained 5-year survival rates as high as 50% [9,10,11,12]; percutaneous ablation gained consensus, as it can grant effectiveness approaching that of surgery in small LM [13]. The treatment of non-colorectal LM is also evolving, but therapies other than chemotherapy are still less codified [14,15,16].

Such an aggressive policy, including several therapeutic options, requires a precision medicine approach. The selection of the appropriate course of action should rely on an adequate understanding of tumor biology and robust clinical biomarkers. However, the availability of reliable prognostic indices is currently an unmet need. Pathologic details of LM can be identified only ex-post after resection. Response of LM to chemotherapy is strongly associated with prognosis [17,18], but it is overestimated by standard imaging modalities [19,20,21]. Genetic mutations are promising biomarkers, but they are still under evaluation [22,23].

In recent decades, we became aware that imaging contains a great amount of data, namely in the form of grey level patterns, which are invisible to the human eye [24]. These texture characteristics can be correlated with pathology data and outcomes [25], potentially allowing diagnostic and prognostic evaluation. The analysis of textural features in medical images, which rely on mathematical functions, such as histogram analysis and matrices, is termed radiomics [24,26]. Recently, radiomic features have been standardized by the imaging biomarker research initiative [27]. This technology is attractive because it could be used to extract biological data directly from the radiological images, without invasive procedures, thus sparing costs and time and avoiding any risk for the patients. It would ideally embody the concept of "virtual biopsy." For many tumors, radiomic analyses have already provided an accurate evaluation of biology, allowing the identification of indices correlated with clinical outcomes [28,29,30,31]. In LM, where multiple therapeutic options are often available, a radiomics-based approach could be used to attain the most appropriate treatment decision. Based on the available literature, the present systematic review aims to elucidate the contribution of radiomic analyses to the management of patients with LM.

## 2. Results

### 2.1. General Characteristics of the Studies

Figure 1 depicts the selection process. After screening for duplicates and eligibility, 32 studies were included in the qualitative synthesis. More than half of the publications (*n* = 18, 56%) were published in the last eighteen months. Most papers (*n* = 28) described retrospective analyses, while four reported planned secondary analyses of prospectively acquired data [32,33,34,35]. Nineteen authors analyzed computed tomography (CT) [32,33,34,35,36,37,38,39,40,41,42,43,44,45,46,47,48,49,50], eight magnetic resonance imaging (MRI) [51,52,53,54,55,56,57,58], three positron-emission tomography (PET)/CT [59,60,61], and two multiple imaging modalities (CT and MRI; PET and MRI, respectively) [62,63]. Various software applications were used for texture analysis, with these being custom-made in a large proportion of cases (*n* = 10).

For the qualitative synthesis, we distinguished four groups of studies according to their subject: (1) radiomics of colorectal LM; (2) radiomics of non-colorectal LM; (3) capability of radiomics to perform differential diagnosis of focal liver lesions, distinguishing LM from other tumors (benign and malignant); (4) technical aspects of radiomics of LM. In the first group (radiomics of colorectal LM), we further distinguished four subgroups of studies according to their endpoints: prediction of survival, prediction of response to chemotherapy, correlation with pathological data, and miscellaneous. For details, refer to Section 4.3. Figure 2 summarizes the organization of qualitative analysis. Most papers (*n* = 18) analyzed radiomics of colorectal metastases. Due to the heterogeneity of studies, some papers fitted into more categories.

### 2.2. Assessment of Study Quality

The average radiomic quality score (RQS) [64] was in the 10 ± 6.5 (range 1–22), roughly 25% of the maximum score (*n* = 36). Only four studies (16%) [33,34,39,40] had a score higher than 18 (>50% of the maximum score). The main limitations in quality were the following: no cost-effectiveness analysis (32 studies, 100%); lack of open-data repositories (*n* = 31, 97%); no phantom calibration (*n* = 31, 97%); failure to include a calibration statistic (*n* = 30, 94%); lack of prospective design (*n* = 28, 87%); and missing validation cohort (*n* = 18, 56%). At the transparent reporting of a multivariable prediction model for individual prognosis or diagnosis (TRIPOD) checklist [65] (31 elements), studies had an average score of 18 ± 3 points (range 14–29), i.e., 58 ± 10% of the maximum possible score. According to the quality assessment of diagnostic accuracy studies (QUADAS-2) [66], there was a high risk of a patient selection bias in 34% of papers because of selection/inclusion criteria in most cases. One-fourth of studies had a high risk of bias related to the index test or the reference standard, while only one study (3%) had a high risk of bias in flow and timing. The RQS and TRIPOD scores of studies are reported in Table 1. Details of QUADAS-2 assessment and summary of its findings are presented in Table 2 and Appendix A. 

Before analyzing the studies’ results in detail, it is helpful to elucidate terminology that is commonly used in radiomics. The definition of radiomic features investigated in the studies is detailed in Table 3. In addition, region of interest (ROI) is defined as the selected area or volume of any imaging modality to analyze for the extraction of radiomic features.

### 2.3. Radiomics of LM from Colorectal Cancer

#### 2.3.1. Prediction of Survival

Ten studies analyzed radiomic features’ ability to predict the outcome of patients with colorectal LM [32,33,38,40,43,46,47,59,61,62]. Four included training and validation datasets [33,37,40,46]. The endpoint was overall survival (OS) in nine studies [32,33,38,40,43,46,47,59,61], progression-free survival (PFS) in four [46,59,61,62], hepatic PFS in one [47], and event-free survival in one [59]. In six papers, patients underwent chemotherapy [32,33,40,43,46,61], in two, liver surgery [47,62], and in two, either chemotherapy or surgery [38,59]. Six studies analyzed imaging modalities before treatment [38,43,47,59,61,62], while four performed comparative analyses of imaging modalities before and after chemotherapy [32,33,40,46]. Two studies performed texture analysis not only of LM but also of non-tumoral liver [38,47]. Finally, six studies analyzed the prognostic role of radiomic features in comparison or combination with traditional biomarkers [33,38,46,47,59,62].

Four studies demonstrated an association between the homogeneity/heterogeneity of LM and survival. Ravanelli et al., reported lower OS and PFS in patients with a higher uniformity of LM at CT scan (cut-off ≥0.42; relative risk (RR) = 6.94; 95% confidence intervals (95%CI) = 1.79–26.79 for OS; RR = 5.05, 95%CI = 1.74–14.66 for PFS) [46]. Andersen et al., described an association between shorter OS and tumor homogeneity at CT (hazard ratio (HR) ranging from 1.5 × 10^20^ to 1.3 × 10^49^, according to the filter used) [32]. Comparing imaging before and after chemotherapy, Dercle et al., identified a radiomic signature associated with OS based on two measures of heterogeneity (spatial heterogeneity and Graytone Difference Matrix contrast, HR = 44.3, 95%CI = 6.4–307.7 for patients with high imaging quality; HR = 6.5, 95%CI = 1.8–23.6 for patients with standard imaging quality) [40]. In the validation setting, the radiomic signature predicted survival better than KRAS-mutational status and 8-week tumor shrinkage evaluated according to RECIST criteria (AUC = 0.80 vs. AUC = 0.67 for KRAS and AUC = 0.75 for RECIST, *p* < 0.001). Finally, in the study by Rahmim et al., at multivariable analysis, LM heterogeneity at ^18^F-FDG PET/CT was a predictor of shorter OS (included into a predictive model, HR = 4.29, 95%CI = 2.15–8.57) [59]. The authors also depicted a model including histogram uniformity, number of metastases, and metabolic tumor volume that was predictive of a shorter event-free survival (HR = 3.20, 95%CI 1.73–5.94, *p* < 0.001) [59].

Three studies showed an association between entropy and prognosis. Andersen et al., and Lubner et al., reported that the higher the entropy of LM, the better the OS (HR ranging from 0.16 to 0.63 in the Andersen et al., study, according to the filter used; HR = 0.65, 95%CI = 0.44–0.95 at coarse filter level in the Lubner et al., study) [32,43]. On the other hand, Beckers et al., reported some prognostic value of the ratio between entropy of LM and entropy of parenchyma (the higher the value, the shorter the OS, HR = 1.9, 95%CI = 0.95–3.78) [38].

Additional radiomic features have been reported. In the Simpson et al., study, LM correlation and contrast (combined into a single texture parameter) were associated with OS (HR = 2.35, 95%CI = 1.21–4.55) [47]. Dohan et al., analyzed imaging modalities before and after treatment and identified three predictors of OS: a decrease in the sum of the target liver lesions, high baseline density of dominant liver lesion, and drop in kurtosis [33]. Those three features (combined into a texture analysis score) evaluated after two months of chemotherapy had a strong association with OS (SPECTRA Score >0.02 vs. ≤0.02, HR = 2.82, 95%CI = 1.85–4.28 in the training dataset; HR = 2.07, 95%CI = 1.34–3.20 in the validation dataset). Radiomic score at two months had the same prognostic value of RECIST criteria after six months of chemotherapy. Shur et al., reported an association of minimal pixel value (negative prognostic factor, HR = 1.66, 95%CI = 1.28–2.16) and gray level size zone matrix (GLSZM) small area emphasis (positive prognostic factor, HR = 0.62, 95%CI = 0.47–0.83) with the PFS [62]. Finally, the following features have been associated with OS: standard deviation [32], LM density at CT scan [46], future liver remnant energy and entropy combined into a single linear predictor [47], ShapeSI4 (included in a radiomic signature) [40], and area under the curve of volume histograms at PET-CT [61].

The results of studies about radiomic features associated with the prediction of survival are summarized in Table 4 and Appendix A.

#### 2.3.2. Prediction of Response to Systemic Chemotherapy

Ten studies analyzed the association between radiomic features and response to chemotherapy [32,33,37,38,40,45,46,54,56,61]. Evidence mainly regarded patients receiving anti-VEGF treatment. Four had training and validation datasets [33,37,40,46]. The reference standard was RECIST criteria in all but one study, which used tumor regression grade (TRG) [45]. Half of the studies analyzed the imaging modalities before and after treatment [32,33,40,45,46], while the other half considered only baseline imaging [37,38,54,56,61]. Four studies focused on targeted therapies associated with systemic chemotherapy [32,33,45,46].

In the Rao et al., paper, the entropy of LM after chemotherapy decreases in responders, while uniformity increases (entropy: −5.13 in responders vs. +1.27 in non-responders, OR = 1.34, 95%CI = 0.92–1.93; uniformity: +30.84 vs. −0.44, respectively, OR = 0.95, 95%CI = 0.89–1.01) [45]. Ravanelli et al., associated a good response with low baseline uniformity (cut-off ≥0.42; OR = 20, 95%CI = 1.85–217.4) [46]. In the study by Beckers et al., treatment success was slightly associated with higher entropy (6.65 ± 0.26 in responders vs. 6.51 ± 0.34 in non-responders, *p* = 0.08) [38]. The Zhang et al., analysis of T2 MRI images before chemotherapy showed that responding lesions had a higher variance and lower angular second moment (two measures of homogeneity) than non-responding ones (variance: 446.07 ± 329.60 in responders vs. 210.23 ± 183.39 in non-responders, *p* < 0.001; angular second moment: 0.96 ± 0.02 vs. 0.98 ± 0.01, respectively, *p* < 0.001) [56]. Dercle et al., built a signature, based on two measures of entropy, gray-tone difference matrix contrast and shape, which allowed to predict responsiveness to anti-angiogenic treatment (AUC = 0.80, CI95% = 0.69–0.94 for patients with high imaging quality; AUC = 0.72, CI95% = 0.59–0.83 for patients with standard imaging quality) [40]. Andersen et al., depicted LM modification after treatment with regorafenib. They observed data discordant with previous analyses (increase in entropy and decrease in uniformity), but none of the patients displayed a measurable response (85% had stable disease, while the remaining ones had progression) [32]. Considering skewness, in the study of Ahn et al., low baseline values (indicating a higher spread towards higher gray levels) were associated with response (0.02 ± 0.32 in responders vs. 0.33 ± 0.44 in non-responders, *p* = 0.001) [37]. One study demonstrated a skewness increase during treatment [32]. In opposition to CT and MRI, high entropy detected at ^18^F-FDG PET images before treatment predicted a worse response to therapy (AUC = 0.74, 95%CI = 0.52–0.97) [61].

Other features have been associated with response: high mean attenuation [37]; narrow standard deviation [37]; high baseline density of dominant liver lesion [33]; and mean values of histogram parameters for apparent diffusion coefficient maps [54].

The results of studies about radiomic features associated with the prediction of response to chemotherapy are summarized in Table 4 and Appendix A.

#### 2.3.3. Prediction of Pathology Data

Three studies evaluated the association between radiomic features and pathology data [39,43,45]. Lubner et al., demonstrated that entropy, mean of positive pixels, and standard deviation are inversely associated with tumor grading (*p* = 0.007 for entropy, *p* = 0.002 for mean positive pixels, and *p* = 0.004 for standard deviation), while skewness and kurtosis showed a trend for an inverse association with KRAS mutation (*p* = 0.04 for skewness, and *p* = 0.058 for kurtosis) [43]. Cheng et al., reported that growth patterns of LM (desmoplastic, replacing, and pushing) can be successfully discerned on CT images by using second-order radiomic features, in particular gray level size zone matrix and gray level non-uniformity (AUC = 0.926, 95%CI = 0.875–0.978 in the training dataset; AUC = 0.939, 95%CI = 0.859–1.000 in the external validation dataset) [39]. In Rao et al.’s paper, the delta in entropy and uniformity values between pre- and post-chemotherapy imaging modalities were predictors of TRG values in patients receiving oxaliplatin-based chemotherapy with or without bevacizumab (entropy variation: −5.13 in TRG 1–2 vs. +1.27 in TRG 3–5, OR = 1.34, CI95% = 0.92–1.93; uniformity variation: +30.84 vs. −0.44, respectively, OR = 0.95, CI95% = 0.89–1.01), while RECIST criteria were not [45].

#### 2.3.4. Other Papers

Three additional papers studied the radiomic features of colorectal LM. Reimer et al., analyzed the evaluation of response of LM undergoing trans-arterial radio-embolization [55]. In post-treatment MRI, higher kurtosis in arterial and portal phases and higher skewness in portal phase identified patients with a progressive disease earlier than standard RECIST criteria. Li Y et al., reported a model based on radiomic features of the primary tumor and LM before resection (heterogeneity, entropy, energy of vertical wavelet, and low-gray-level run emphasis) that was able to predict the future appearance of further LM [42]. Wagner et al., analyzed CT and PET-CT imaging of primary tumor and LM [60]. They demonstrated that colon cancer and LM have different skewness and kurtosis at both imaging modalities (CT and PET-CT), while colon cancers with or without LM have similar features.

Table 4 summarizes the data of studies dealing with colorectal LM.

### 2.4. Radiomics of Non-Colorectal LM

Four papers focused on non-colorectal LM. A single study assessed CT-based radiomic indices in LM from esophageal cancer [41]. The study found that the characteristics of pre-treatment CT related to heterogeneity and gray-level intensity, such as wavelet gray level co-occurrence matrix correlation and gray level distance zone matrix with large dependence emphasis, were predictors of response to chemotherapy. Two studies explored radiomic analyses in LM from neuroendocrine tumors (NET) [44,63]. Martini et al., analyzed a small series of patients (*n* = 49) but observed a number of associations: pancreatic NET had lower skewness and higher mean HU than non-pancreatic ones; entropy in the arterial phase was negatively associated with PFS in pancreatic NET and with OS in non-pancreatic NET; kurtosis was associated with lower OS in pancreatic NET, while skewness with higher one [44]. Weber et al., investigated the correlation between parameters derived from the somatostatin receptor agonist (^68^Ga-DOTATOC) PET and MRI with the proliferation index Ki67 [63]. Entropy and dissimilarity (from both PET and MRI) had a direct correlation with Ki67, while homogeneity had an inverse one. Moreover, it was possible to distinguish G1 and G2 LM on the basis of entropy, homogeneity, and dissimilarity (on PET data only). Finally, Trebeschi et al., reported heterogeneity-related radiomics parameters as predictors of response to immunotherapy in LM of melanoma and non-small-cell lung carcinoma [49].

### 2.5. Differentiation of LM from Other Hepatic Lesions

Four studies investigated whether radiomic features could discriminate LM from other hepatic lesions. Jansen et al., analyzed metastases, primary hepatic tumors, and benign lesions (adenomas, cysts, and hemangiomas) on MRI images [52]. A model using, among other features, the time to peak histograms and the sum of squared variance could distinguish different liver lesions. In the paper by Gatos et al., selected texture characteristics (inverse different moment, sum variance, and long-run emphasis) could differentiate metastases, hepatocellular carcinoma (HCC), and benign lesions [51]. Li et al., tested a model to distinguish hemangiomas, LM, and HCC, using second-order features (gray level co-occurrence matrix, gray-level run-length matrix, and intensity-size zone matrix) [53]. In their model, no feature combination could differentiate the three types of lesion at the same time. Differential diagnosis of the two malignant entities (LM and HCC) required a more complex model, with a higher number of features, than differential diagnosis between benign and malignant lesions (LM vs. hemangiomas or HCC vs. hemangiomas). Finally, a study by Song et al., identified kurtosis, variance, and inverse difference moment as distinguishing criteria between benign and malignant hypervascular lesions [48]. Only the latter study used pathology data as the reference standard for all the analyzed patients.

Figure 3 provides an overview of the potential contribution of radiomics to the management of patients with LM.

### 2.6. Influence of Technical Features on Radiomic Analyses

Some studies set off to investigate whether acquisition or reconstruction parameters could influence the values of texture analysis indices. Ahn et al., tested three different reconstruction modes of CT images, i.e., filtered back-projection, iterative reconstruction model, and hybrid iterative reconstruction. The reconstruction method affected numerous parameters, including entropy, homogeneity, skewness, kurtosis, and gray-level co-occurrence matrices [36]. Lubner et al., compared the effect of 2D and 3D reconstruction on radiomic parameters by performing a Bland–Altman analysis on a subset of 20 patients [43]. The results were similar for the two methods. Those results were confirmed by a further investigation by Ahn et al. [37]. The latter study also compared the influence of different CT scanners, ranging from 8 to 64 rows, on the radiomic parameters, without finding any significant difference. Similar results were reported in the MRI setting: Peerlings et al., used the concordance correlation coefficient to test the reproducibility of an array of first- and second-level radiomics parameters over time (multiple MRI) and different MRI systems on apparent diffusion coefficient maps, finding good stability with most parameters [58]. Conversely, two studies reported that radiomic parameters derived from CT scans are affected by slice thickness setting and ROI size [34,50]. Dercle et al., demonstrated that ROI area size, metastatic site, and the individual characteristics of image acquisitions should be considered as confounding factors in the evaluation of tumor entropy [35].

Inter-observer agreement was assessed by four studies [33,52,54,60]. Although they used different indices, such as K-statistic, intra-class correlation, and correlation index (*r*-value), all studies reported a substantial or excellent agreement among different readers. Finally, one study by Chatterjee et al., devised a method to reduce the rate of false discovery when analyzing radiomic parameters in small datasets [57].

## 3. Discussion

Our review identified a consistent number of research papers dealing with radiomics analyses of LM, mostly published in the last three years and with an evident increase over time. The role of texture analysis has been explored in different clinical settings, leading to innovative insights into the management of patients with LM.

In the diagnostic field of research, radiomics can distinguish different types of hepatic lesions, differentiating metastases from benign lesions and primary tumors [48,51,52,53]. LM appear to be characterized by a high gray level entropy, heterogeneity, and variance. This phenomenon can be explained by the co-existence of different cell clones, the presence of necrosis and, more relevantly, by the unregulated sprout of new tumor vasculature. Radiomic analyses could lead to a conclusive and reliable diagnosis after a single imaging modality, preventing the need for an additional radiological examination or an invasive biopsy.

Most studies focused on LM from colorectal cancer. Higher entropy and lower homogeneity of LM at diagnosis have been associated with a better prognosis and response to therapy [38,40,45,46,56]. Such indices could predict good vascularization of LM, while more homogeneous tumors could reflect a tighter cellular structure or necrosis, which might purport reduced therapy effectiveness. Conversely, a decrease in entropy and an increase in homogeneity after treatment on CT have been associated with tumor response [40,45]. Similar considerations are possible for skewness: low baseline values and an increase after chemotherapy were associated with response to chemotherapy [32,37]. A higher asymmetry index describes a higher prevalence of voxel with lower gray level values, which is compatible with the onset of necrosis in the target tissue. All those radiomic features are consistent with the reduction in neoangiogenesis and the onset of necrosis. It is worth noting that measures of heterogeneity could have a different meaning in ^18^F-FDG PET-CT [59,61], indicating therapy resistance. Further studies are needed for this imaging modality.

Some studies, such as the one by Rao et al. [45], demonstrated the superiority of radiomic features over standard biomarkers and predictors of response to chemotherapy. Such data are of major clinical relevance considering that, to date, prognosis assessment in patients with colorectal LM is still limited: it relies on morphological criteria, while tumor biology assessment by genetic factors is largely unsatisfactory [22,23]. Similarly, traditional RECIST criteria for response evaluation are associated with prognosis [17,18] but show major discrepancies with real LM modifications at the pathological level (TRG) [19,20]. Radiomic features not only demonstrated an earlier evaluation of response than standard RECIST criteria [33], but also an adequate assessment of TRG [45]. Texture analyses were also able to predict additional pathology details of LM that have a prognostic impact, such as tumor growth patterns [39]. Those data can allow a real precision medicine, planning treatment based on a reliable evaluation of the effectiveness of therapies and prognosis, that, to date, can be assessed only ex-post. However, when evaluating the radiomic data, it is of utmost importance to place the texture information into the appropriate clinical context. For instance, indices related to uniformity are negative biomarkers at baseline, but hallmarks of good response in the post-treatment setting.

Evidence was much less robust for non-colorectal LM. The predictive value of heterogeneity at baseline staging is shared by LM from colorectal and esophageal cancer, but not by NET [44]. In the latter group, the possibility of identifying the origin and grading of NET through radiomics is appealing [44], since both data drive treatment and prognosis. Nonetheless, data are still limited and must be confirmed by further studies.

The present review highlights some significant limitations of studies, as reflected by the methodological RQS assessment and the clinical TRIPOD checklist. The selected papers presented a wide variability in sample size, relying on small series in most cases. Only a few studies had a prospective design (radiomic analyses being a secondary endpoint in all of them) or a validation dataset. Even comparison with reference standard was not adopted by all authors. There was a high heterogeneity of utilized techniques and inconsistencies in the number and order of analyzed features, as almost every institution used a different software application. Furthermore, most studies did not provide a univocal cut-off value of radiomic features or, when provided, it was not coherent among studies, precluding any broader applicability of mentioned parameters. To date, only a few studies considered data from surgical specimen or biopsy. This is a major limitation as long as pathological data are the mandatory reference to definitively assess the capability of radiomics to discriminate LM from other diseases and to identify the biological characteristics of tumors.

All these limitations hinder the quantum leap of radiomics from the investigation field into clinical practice, representing a central issue of future radiomics research, and should be addressed as soon as possible. The instability of radiomic features across different devices and acquisition protocols, especially for MRI images [67], could further limit the real application of radiomics to daily clinical practice. To this purpose, a collaboration between clinicians and medical imaging experts is pivotal, as interdisciplinarity correlates with the quality of the published research [68]. Cooperation between institutions is warranted to find methods capable of countering features variability and instability, based on the analysis of large databases [69]. Continuous standardization of radiomic features is also crucial [27]. The radiomic analysis should be performed with a validated and user-friendly software interface, as the heterogeneity of methods prevents the attainment of reproducible cut-off values [70]. Transition to clinical practice is impossible without a reliable and fast segmentation tool, which must be able to identify and isolate the target structure semi-automatically. In this setting, an adaptive threshold could be applied [71]. Machine learning methods, such as convoluted neural networks, in combination with radiomics, appear particularly promising, especially for the identification and segmentation of small lesions [26,72,73]. To date, no study has pursued this approach for LM.

These limitations notwithstanding, some data are encouraging. Independently of the adopted methodology, studies addressing similar questions came to similar conclusions. Analogously, radiomic parameters relevant for a given clinical situation were reproducible across studies. Different analysis techniques, such as 3D or 2D feature extraction, did not have a relevant impact on the obtained values. Likewise, using different scanning devices or switching the operator performing the analysis did not affect the information’s reproducibility. Finally, even imaging modalities based on entirely different physical principles (magnetic resonance and CT) yielded similar results in some settings.

Some limitations of the present review could be argued. The study was designed according to a wide scientific question rather than according to specific PICO questions. This is a first explorative review about the role of radiomics in LM, for which available studies were expected not to provide high-level evidence. We aimed to give an overview of a cutting-edge topic. The extreme heterogeneity of imaging modalities and software packages used, of clinical scenarios (LM from different tumors, patients with/without chemotherapy, data before/during/after systemic or loco-regional therapies), and of clinical endpoints (diagnosis, prognosis, and effectiveness of treatment) precluded the possibility of performing a meta-analysis of data. We did not consider ultrasonography despite its wide diffusion for liver tumors. In fact, the operator-dependent origin of image data would have carried a relevant risk of bias. As mentioned, we did not include machine learning methods, but, to date, no study has used such combination of artificial intelligence and texture analysis for LM. In such a rapidly evolving field, these limitations should be insights for future research perspectives.

## 4. Materials and Methods

### 4.1. Database Search Strategy

We performed a systematic search of PubMed, Science Citation Index, Embase, and clinicaltrial.gov databases and web sources (Google Scholar) for articles relevant to radiomics of LM. The adherence of the present review to PRISMA guidelines was assessed by the PRISMA checklist (Appendix A). We decided not to formulate PICO questions because of the expected heterogeneity of studies and the low level of available evidence. The study was registered on the PROSPERO database (CRD42020193930) at the end of the analysis.

The search algorithm was constructed using the following terms: “radiomics” OR “texture analysis” OR “radiological features” OR “radiomic features” OR “textural features” AND “hepatic metastases” OR “liver metastases”. Only full-text articles in English, reporting on human subjects, written and published (including those distributed as “online first”) as of May the 31st, 2020, were considered. The search was then expanded by reviewing the reference list of the selected articles. Two authors (F.F. and N.G.) reviewed each manuscript and eliminated those not fitting the inclusion criteria (detailed below); in cases of discordance, a consensus was reached after discussion with an independent author (L.V.).

### 4.2. Study Selection and Quality Appraisal

Studies reporting feature extraction from diagnostic images in patients affected by LM were included in the present review. Studies describing analyses of purely semantic (visual qualitative) features, such as size, lobulation, spiculation, and radiological signs of vascular invasion, were not included. Papers describing computer-assisted tumor recognition, such as convoluted neural networks, were included only if at least one textural feature was used in the process. No study was excluded because of sample size (except for case reports). All tomographic radiological and nuclear medicine modalities were allowed: this included contrast-enhanced and unenhanced CT and MRI, as well as PET/CT with any tracer. The following papers were excluded:Articles not matching the field of interest of the current review.Other review articles (however, these articles were screened for references).Editorial, letters, or conference proceedings.Reports of single cases.Reports on ultrasound imaging or other operator-dependent technique.Phantom, simulation or small animal studies.

In the first step of the selection process, the article title and abstract were screened; whenever ineligible, according to the aforementioned criteria, the article was omitted. In the second step, the full text of the articles was assessed to determine paper eligibility. In the case of positive evaluation, the entire reference list was manually examined to detect other potential candidate articles, which might have been left out by the search algorithm. The quality of the included studies was assessed by using the methods-related Radiometrics Qualitative Score, as proposed by Lambin et al. [64], and the clinically-oriented TRIPOD checklist, as proposed by Park et al. [65]. The presence of relevant bias in the included studies was evaluated according to QUADAS-2 [66]. Two readers (F.F. and N.G.) evaluated the scores, with a third senior reader (L.V.) being referred to whenever a consensus was needed.

### 4.3. Articles and Features Classification

For each article that passed the selection process, the following data were extracted and organized in a table: basic article metrics, including name of the first author, institution of the corresponding author, journal, and year of publication. Then, information related to the study design, type of primary tumor, target of analysis (LM only or LM and healthy liver parenchyma/primary tumor), endpoints, sample size, and radiological technique were inserted. Considering the study endpoints, data about survival, response to chemotherapy, pathological details, and technical issues were collected. The term “survival” includes overall survival, progression-free survival, event-free survival, and recurrence-free survival. The radiological and the pathological assessment of response to chemotherapy were considered separately because of the discrepancy that may occur between the two [19,20]. Pathological data include tumor characteristics (e.g., grading and growth pattern), and genetic mutations. Finally, data concerning the software package used to carry out the analysis and the radiomic features extracted were collected. Textural features included descriptors of the voxel distribution curve (mean, skewness, and kurtosis), of the homogeneity of the intensity values (energy, entropy, angular second moment), of the frequency of adjacent voxels with the same values (gray level co-occurrence matrices, gray level run length matrices, and gray level size-zone matrices), and, finally, the intensity difference between voxels (neighboring gray level difference matrices). Table 3 provides an overview of the most common radiomic features.

## 5. Conclusions

A number of clinical messages can already be extrapolated. Radiomics allow non-invasive differential diagnosis of hepatic lesions. More importantly, radiomic characteristics can foretell the outcome of patients, and the therapeutic effectiveness of treatments, outperforming standard predictive and prognostic models (Figure 3). Altogether, radiomics has the potential to offer a significant contribution to the precision medicine approach. However, interdisciplinarity, standardization, and reproducible software applications are indispensable tools for the transition of radiomics into clinical practice.

## Figures and Tables

**Figure 1 cancers-12-02881-f001:**
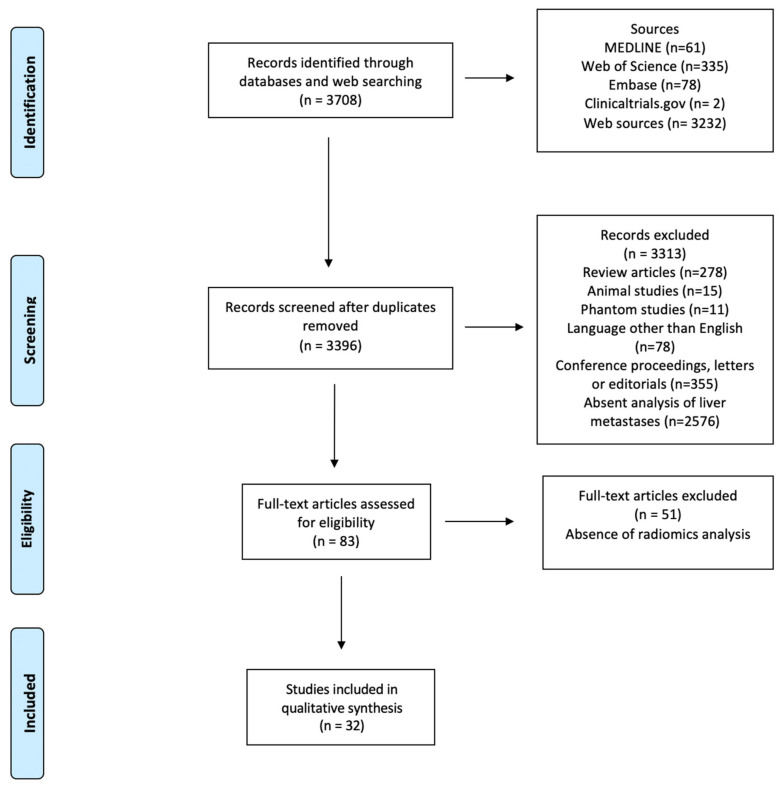
PRISMA Flowchart of study selection.

**Figure 2 cancers-12-02881-f002:**
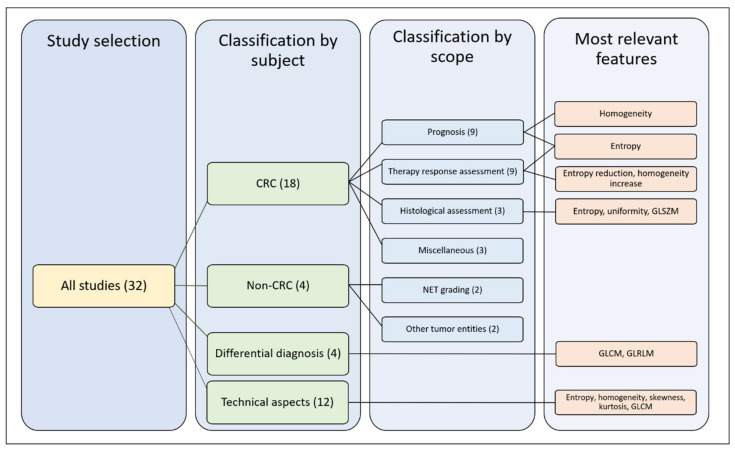
Outline of the selected studies organization. CRC: colorectal cancer, GLSZM: gray level size zone matrices, GLCM: gray level co-occurrence matrices, GLRLM: gray-level run-length matrices.

**Figure 3 cancers-12-02881-f003:**
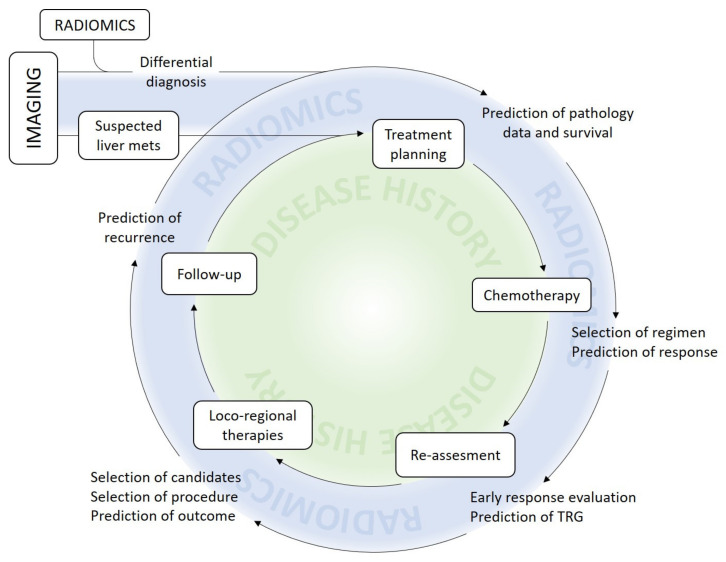
Present and potential future contribution of radiomics to clinical practice.

**Table 1 cancers-12-02881-t001:** Details of studies included in the review.

First Author	Year	Diagnosis	#	Radiological Technique	Analyzed Series	Radiomics Software Program	Analysis of Second-Order Radiomic Features	Relevant Radiomics Features	RQS (%)	TRIPOD (%)
Ahn S.J. [37]	2016	CRC	235	CT	PVP	C++ based	N	Skewness	4 (11%)	18 (58%)
Ahn S.J. [36]	2019	NS	259	CT	PVP	Custom C++	Y	Skewness, kurtosis, entropy, uniformity, and GLCM	4 (11%)	20 (65%)
Andersen I.R. [32]	2019	CRC	27	CT	Dynamic contrast, PVP	MatLab-based	N	Skewness, entropy, and uniformity	12 (33%)	19 (61%)
Beckers R.C.J. [38]	2018	CRC	70	CT	PVP	MatLab-based	N	Entropy and uniformity	5 (14%)	21 (68%)
Chatterjee A. [57]	2018	NS	69	MRI	T1, Fast spin-echo T2, DWI	NS	Y	GLCM, GLRLM	17 (47%)	15 (48%)
Cheng J. [39]	2019	CRC	94	CT	Arterial, PVP	NS	Y	GLSZM, GLNU	21 (58%)	20 (65%)
Dercle L. [40]	2020	CRC	667	CT	PVP	NS	Y	Entropy, GTDM, Shape	22 (61%)	13 (42%)
Dercle L. [35]	2017	NS	14	CT	PVP	TexRAD	N	Entropy	11 (31%)	22 (71%)
Dohan A. [33]	2019	CRC	230	CT	PVP	TexRAD	N	Kurtosis	21 (58%)	20 (65%)
Gatos I. [51]	2017	Multiple	22	MRI 1,5 T	T2- and DWI	NS	Y	GLCM, GLRLM	7 (19%)	14 (14%)
Jansen M.J.A. [52]	2019	Multiple	95	MRI 1,5 T	T2-weighted	NS	Y	Entropy, GLCM	15 (42%)	16 (52%)
Klaassen R. [41]	2018	EC	18	CT	Late contrast phase	MatLab-based	Y	GLCM	14 (39%)	23 (74%)
Li Y. [42]	2019	CRC	24	CT	PVP	ITK-SNAP	Y	Entropy, uniformity, and GLRLM	15 (42%)	18 (58%)
LI Z. [53]	2017	NS	67	MRI 3 T	T2-weighted SPAIR	NS	Y	GLCM, GLRLM, GLSZM	14 (39%)	17 (55%)
Liang H.Y. [54]	2016	CRC	53	MRI 1,5 T	ADC maps, arterial, PVP	MaZda	N	None significant	2 (5%)	16 (52%)
Lubner M.G. [43]	2015	CRC	77	CT	PVP	TexRAD	N	Entropy, Energy	5 (14%)	18 (58%)
Martini I. [44]	2019	NET	48	CT	Arterial and PVP	TexRAD	N	Skewness, Kurtosis, and Entropy	1 (3%)	16 (52%)
Meyer M. [34]	2019	CRC	78	CT	PVP	Radiomics	Y	Kurtosis, energy, GLCM, GLRLM, and GLSZM	20 (56%)	16 (52%)
Peerlings [58]	2019	Multiple	30	MRI 1,5 and 3 T	DWI	MatLab-based	Y	GLCM, GLRLM, GTDM, GLSZM	6 (16%)	18 (58%)
Rahmim A. [59]	2019	CRC	52	PET/CT	18F-FDG PET	NS	N	Uniformity	6 (16%)	19 (61%)
Rao S.X. [45]	2015	CRC	21	CT	PVP	MatLab-based	N	Entropy and uniformity	1 (3%)	19 (61%)
Ravanelli M. [46]	2019	CRC	43	CT	PVP	MatLab-based	N	Uniformity	16 (45%)	17 (55%)
Reimer R.P. [55]	2018	Multiple	37	MRI 1,5 T	T1, PVP and hepatocellular	Mint Lesion	N	Skewness and Kurtosis	5 (14%)	17 (55%)
Shur J. [62]	2019	CRC	102	CT; 1,5/3 T MRI	PVP (CT); T1 FS and hepatocellular MRI	Pyradiomics	Y	GLSZM	7 (19%)	29 (94%)
Simpson A.L. [47]	2017	CRC	198	CT	PVP	NS	Y	GLCM	5 (14%)	19 (61%)
Song S. [48]	2019	Multiple	20	CT	Arterial phase	Omni-Kinetic	Y	Kurtosis, GLCM	15 (42%)	15 (48%)
Trebeschi [49]	2019	Multiple	NS	CT	NS	NS	Y	GLSZM	17 (47%)	16 (52%)
Van Helden E.J. [61]	2018	CRC	47	PET/CT	18F-FDG PET	NS		Entropy and Shape	8 (25%)	17 (55%)
Velichko Y.S. [50]	2020	BC	54	CT	PVP	LIFEX	Y	Uniformity and GLCM	5 (14%)	15 (48°%)
Wagner F. [60]	2017	CRC	18	CT; PET/CT	PVP (CT), 18F-FDG PET	Pmod 3.5	N	Skewness and kurtosis	1 (3%)	19 (61%)
Weber M. [63]	2019	NET	100	PET/MRI	68Ga-DOTAPET; MRI ADC	LIFEX	Y	Entropy, uniformity, and GLCM	5 (14%)	16 (52%)
Zhang H. [56]	2018	CRC	26	MRI 3 T	T2-weighted	MatLab-based	Y	GLCM	9 (25%)	17 (55%)

CRC: Colorectal Cancer, NET: Neuroendocrine Tumor, EC: esophageal cancer, BC: breast cancer, PVP: portal venous phase, FS: fat suppression, DWI: diffusion-weighted images, SPAIR: spectral-attenuated inversion recovery, ADC: apparent diffusion coefficient, GLCM: gray level co-occurrence matrix, GLRLM: gray-level run-length matrix, GTDM: gray-tone difference matrix, GLSZM: gray level size zone matrix, GLNU: gray-level non-uniformity, NS: not specified.

**Table 2 cancers-12-02881-t002:** QUADAS-2 evaluation of studies.

Evaluation	Risk of Bias				Applicability Concerns		
	Patient Selection	Index Test	Reference Standard	Flow and Timing	Patient Selection	Index Test	Reference Standard
Low Risk	15 (47%)	18 (56%)	18 (56%)	28 (87%)	21 (65%)	21 (65%)	19 (59%)
High Risk	11 (34%)	8 (25%)	8 (25%)	1 (3%)	8 (25%)	6 (19%)	6 (19%)
Unclear	6 (19%)	6 (19%)	6 (19%)	3 (10%)	3 (10%)	5 (16%)	7 (22%)

**Table 3 cancers-12-02881-t003:** Overview of the analyzed radiomic characteristics.

Feature Family	General Descriptio	Common Features	Specific Description
Shape & Size	2D/3D geometric properties of the region of interest	Area	
		Volume	Number of voxels in the ROI
		Maximum 3D diameter	The maximum distance between any two voxels on the surface of the ROI
		Major axis length	
		Minor axis length	
		Surface Area	
		Compactness	How compact the region is independent of scale and orientation
		Elongation	The inverse of compactness
		Flatness	Absence of curvature in an ROI
		Sphericity	The roundness of the shape of the ROI relative to a sphere
		Spherical disproportion	ROI surface area/surface area of a sphere based on ROI radius
First Order Statistics	Intensity distribution in the ROI based on the intensity histogram, regardless of the spatial relationships	Intensity	Includes mean, min, max, SD, and percentiles
		Energy	The magnitude of voxel intensities (sum of square voxel values)
		Uniformity/homogeneity	The sum of the squares of each intensity value in the ROI
		Entropy	Amount of information in voxel values
		Skewness	Asymmetry of values. Negative skewness: data are skewness to the right of the mean (higher values). Positive skewness: data are skewed to the left of the mean (lower values).
		Kurtosis	Distribution of voxel values. Low kurtosis: most data points are close to the mean (few outliers). High kurtosis: data are spread far from the mean (more outliers).
Second-Order Statistics	Textural features quantifying tumor heterogeneity by analyzing the spatial distribution of pixel/voxel intensities	Gray level co-occurrence matrix (GLC M)	Measures the arrangements of voxel pairs along a fixed direction (homogeneity, contrast, correlation, entropy, dissimilarity, and angular second moment/energy)
		Gray level run length matrix (GLRLM)	Consecutive voxels with the same intensity along with fixed directions (can have long- or short-run, as well as low- and high-gray level emphasis).
		Gray level size zone matrix (GLSZM)	Clusters of connected pixels with the same grey value. They can have small- and large-area as well as low- or high-gray emphasis.
		Neighborhood gray tone/level difference matrix (NGTDM/NGLDM)	The difference in gray level between one voxel and its 8/26directions (in 2D/3D). Includes rate, intensity, and frequency of intensity change.
Model-or transform-based Statistics	Relationship between three or more pixels or voxels	Autoregressive model	Filters or mathematical transforms to the images identifying repetitive or non-repetitive patterns, suppressing noise, or highlighting details, extract areas with increasingly coarse texture patterns
		Wavelet transform	
		Fractal analysis	
		Minkowski functionals	
		Fourier transform	

**Table 4 cancers-12-02881-t004:** Studies on radiomics in patients with liver metastases from colorectal cancer.

First Author	Year	Design	#	Imaging	Analyzed Imaging	Main Intervention	Intervention Type	Pathology Data	Validation Cohort	Outcome Measures	Synopsis of Findings
Survival											
Lubner M.G. [43]	2015	R	77	CT	Pre-therapy	Systemic therapy	NS	Y	N	OS	Entropy correlated with longer OS
Simpson A.L. [47]	2017	R	198	CT	Pre-therapy	Surgery	Metastasectomy	N	N	OS/HDFS	Tumor correlation and contrast were associated with OS; future liver remnant energy and entropy were associated with OS and HDFS
Andersen I.R. [32]	2019	P	27	CT	Pre/post-therapy	Systemic therapy	Regorafenib	N	N	OS	LM uniformity predicted shorter OS; LM entropy predicted longer OS
Beckers R.C.J. [38]	2018	R	70	CT	Pre-therapy	Systemic therapy or surgery	XELOX with or w/o Bevacizumab	N	N	OS	LM/parenchyma entropy ratio correlated with disease burden and OS.
Dercle L. [40]	2020	R	667	CT	Pre/post-therapy	Systemic therapy	FOLFIRI with-w/o Cetuximab	N	Y	OS	A signature using Shape SI4, Log Z/X Entropy, GTDM Contrast can predict OS.
Dohan A. [33]	2019	P	230	CT	Pre/post-therapy	Systemic therapy	FOLFIRI and Bevacizumab	N	Y	OS	A radiomic score granted earlier OS prediction than RECIST 1.1
Rahmim A. [59]	2019	R	52	FDG PET	Pre-therapy	Mixed	Systemic, RF, or Metastasectomy	N	N	OS, EFS	LM heterogeneity predicts OS; histogram uniformity predicts EFS
Ravanelli M. [46]	2019	R	43	CT	Pre/post-therapy	Systemic therapy	FOLFIRI/FOLFOX with-w/o Bevacizumab	N	N	OS, PFS	Uniformity was related to PFS; uniformity and CT density were associated with OS in the EGFR group
Shur J. [62]	2019	R	102	CT; MRI	Pre-surgery	Neoadjuvant therapy and surgery	NS, Metastasectomy	N	N	DFS	Minimum pixel value and GLSZM small area emphasis are associated with DFS.
Van Helden E.J. [61]	2018	R	47	FDG PET	Pre-therapy	Systemic therapy	XELOX with or w/o Bevacizumab or Cetuximab	N	N	OS, PFS	AUC-ISH predicted OS and PFS
Response to Chemotherapy											
Ahn S.J. [36]	2016	R	235	CT	Pre-therapy	Systemic therapy	FOLFIRI or FOLFOX	N	Y	RECIST	Lower skewness on 2D, higher attenuation in 3D, narrower SD on 3D predict the therapy response
Andersen I.R. [32]	2019	P	27	CT	Pre/post-therapy	Systemic therapy	Regorafenib	N	N	RECIST	Entropy and skewness increased; uniformity decreased after treatment
Beckers R.C.J. [38]	2018	R	56	CT	Pre-therapy	Systemic therapy	XELOX with-w/o Bevacizumab	N	N	RECIST	LM entropy showed a trend for being higher in responders
Dercle L. [40]	2020	R	667	CT	Pre/post-therapy	Systemic therapy	FOLFIRI with-w/o Cetuximab	N	Y	RECIST	Shape SI4, Log Z/X Entropy, GTDM Contrast can predict Cetuximab sensitivity.
Dohan A. [33]	2019	P	230	CT	Pre/post-therapy	Systemic therapy	FOLFIRI and Bevacizumab	N	Y	RECIST	LM density, integrated into a radiomics score, identified responders.
Liang H.Y. [54]	2015	R	53	MRI	Pre-therapy	Systemic therapy	Fluorouracil-based chemotherapy	N	N	RECIST	Mean ADC values are lower in responders.
Rao S.X. [45]	2015	R	21	CT	Pre/post-therapy	Systemic therapy	XELOX with-w/o Bevacizumab	Y	N	TRG	A decrease in entropy and uniformity increase after treatment correlates with response.
Ravanelli M. [46]	2019	R	43	CT	Pre/post-therapy	Systemic therapy	FOLFIRI/FOX with-w/o Bevacizumab	N	N	RECIST	Uniformity discriminated EGFR responders from non-responders
Van Helden E.J. [61]	2018	R	47	FDG PET	Pre-therapy	Systemic therapy	XELOX with or w/o Bevacizumab or Cetuximab	N	N	RECIST	Entropy was higher in patient non-responders
Zhang H. [56]	2018	R	26	MRI	Pre-therapy	Systemic therapy	FOLFORI or FOLFOX or XELOX	N	N	Size change	Responding LM had a higher variance and lower angular second moment
Miscellaneous											
Cheng J. [39]	2019	R	94	CT	Pre-therapy	Surgery	Partial hepatectomy	Y	Y	HGP	A clinic-radiomics model (GLSZM and gray level non-uniformity) can predict growth patterns.
Li Y. [42]	2019	R	24	CT	Pre-therapy	Surgery	Colectomy, lymphadenectomy, metastasectomy	N	Y	LM occurrence	Heterogeneity, entropy, energy, and GLRLM_LGE predicted the risk of LM
Reimer R.P. [55]	2018	R	16	MRI	Post-therapy	TARE	TARE with 90Y-microspheres	N	N	RECIST	High kurtosis (arterial/venous) and low skewness (venous) identified progression
Wagner F. [60]	2017	R	18	CT; FDG PET	Pre-therapy	Systemic therapy	NS	N	N	Primary/LM	Skewness and kurtosis (CT) and kurtosis (PET) are different in primary and LM

R: Retrospective, P: Prospective, CRC: Colorectal Cancer, NET: Neuroendocrine Tumor, LM: liver metastases, RF: radiofrequency, XELOX: Capecitabine and Oxaliplatin, FOLFIRI: Fluorouracil and Irinotecan, FOLFOX: Fluorouracil and Oxaliplatin, TARE: trans-arterial radioembolization, OS: overall survival, HDFS: hepatic disease-free survival, EFS: event-free survival, PFS: progression-free survival, RECIST: response evaluation criteria in solid tumors, HGP: histological growth patterns, GLSZM: gray level size zone matrix, GTDM: gray-tone difference matrix, ADC: apparent diffusion coefficient, AUC-ISH: area-under-the-curve of cumulative SUV/Volume histograms, NS: not specified.

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
