# Peer review of "Radiomics of Liver Metastases: A Systematic Review"

_cancers, 2020, doi:10.3390/cancers12102881_

Round 1

Reviewer 1 Report

Thank you for the opportunity to review this extensive review.

The authors give a relevant overview about the published literature of Radiomics analyses in liver metastasis.

It is a very important field of imaging research and a good overview of the literature is needed.

There are different methodical approaches due to imaging modality and software used, different clinical questions, including prognosis and treatment prediction.

Therefore, no meta analysis can be performed from the literature and only the presented systematic review can be given.

The authors used the suggested Radiomics quality score to assess the papers.

I would like to see the QUADAS2-assessment to test for possible bias of the studies.

Reviewer 2 Report

This is a very comprehensive review on radiomics of liver metastasis.  It analyzes the relevant literature and makes appropriate conclusions.  I only have several minor concerns:

  1.  Ultimately, tissue obtained from a biopsy provides the definitive diagnosis.  While imaging is helpful, and for HCC often diagnostic, biopsy may still be necessary especially when dealing with a tumor of unknown origin.  These imaging characteristics used in radiomics may predict a diagnosis, however, concordance with a biopsy is important to determine the sensitivity and specificity of these imaging traits. Comment in the manuscript  on radiomics and the gold standard biopsy will be helpful.   
  2. Also, several terms are used in the manuscript such as entropy, homogeneity, etc; it would help the general reader if these terms are defined.

Reviewer 3 Report

Paper description

Fiz et al performed a systematic review of literature in major databases regarding the a radiomics based approach to evaluating liver metastases without an explicitly clarified scientific question. They then systematically outline the different imaging features and editorialized their relative utility. PRISMA guidelines were purportedly used, meta-analysis was not enacted.

Summary

The intent of this work has merit and would be readily worthy of publication on scientific merit. I believe it would also be cited as it addresses a poorly indexed question in Medline and there seems to be a thorough approach to sharing the relevant data in the literature.

There is a key issue in the methodology - a large amount of literature seems to have been missed by only considering the term "metastasis" when my quick search to better understand why no included literature implemented CNNs found that the literature frequently used the terms "masses" or compared metastases to HCC. Since these scenarios fall within the scope of the implied question of the authors (i.e. can we distinguish liver mets from HCC), there is an issue. This may not have been the authors' intention, but there is not a clear scientific question stated in the Abstract nor Methods  ("We perofrmed a systematic reviwe of literature about radiomic analyses in patients with LM…")

Elsewhere, repeated small issues in the writing make the quality of the work unclear and thus circumspect. This is likely a minor translation issue that has become a large issue due to their multitude in subtle fashions - my direct suggestions later are not exhaustive and a thorough review is required. It does feel that the written results section is editorialized secondary to the absence of a Table directly comparing the predictive capacity of different features nor the presentation of metrics of error.

Minor revisions must be considered to clarify the methodology and results as well as to minimize overly simplistic generalizations. I would still hold reservations based on the apparent absence to clearly set their methods a priori or show the supplemental work to support this (ex.  a table outlining which studies were included and excluded). Please note that the PROSPERO protocol was published months after the search date was finalized

In truth, if the authors did not observe a standard systematic format (ex. PRISMA) this should be highlighted as a major limitation and I would encourage publication regardless, following the aforementioned Minor Revisions. If the authors did observe a true systematic format (ex. PRISMA in its entirety), then this should be emphasized in the abstract. Right now, the absence of a PRISMA checklist and the other requisite items in it make it appear this methodology may not have been observed. I recognize that their methods state it was implemented and the registration of a protocol is IMMENSELY reassuring that this is all just an issue with communication.

In my opinion, this paper should be refused if the authors do not agree to these clarifications and a thorough attempt at these revisions - I would encourage a rejection if the revisions were not approached in good faith. The missing search Hedge is not necessary - it could be clearly stated as a limitation. However, these items must be clearly highlighted as limitations.

Strengths

  • Relevant summary of a poorly indexed body of literature - conventional Medline searches generated far fewer search returns
  • Very relevant summary for those considering feature studies in this area
  • Approachable writing style for clinical readership
  • Registered protocol

Limitations

  • The abstract does not communicate adequate nor correct information to summarize their work - in particular, the single line describing their methods lacks relevant details implying that an accepted systematic format was not observed
  • Unclear scientific question in Abstract and Methodology section
    • Their PROSPERO protocol states this very clearly however
  • Registration of PROSPERO protocol on July 24th, 2020, after May 2020 search termination, is suspicious for registering the a priori protocol following a review of the literature
  • Repeated small grammar issues result in terms being used that are not consistent with the clinical or radiomics literature, these are benign but confusing (ex. Repeated use of the term prognosticator when biomarker is more appropriate; repeatedly implying that you foretell a prognosis, however the prognosis is the foretelling of the survival - you are adding a needless instrument in this illutration; MR instead of MRI; VOI instead of ROI - both terms are used in the paper….)
  • Tone at some points makes the apparent synthesis of the facts appear as editorialized rather than unbiased
  • No documented reason to not proceed to meta-analysis
  • No supporting documentation to support that PRISMA structure was observed
  • Missed a seemingly relevant hedge of the use of convolutional neural networks to distinguish the etiology of liver masses (ex. https://www.ncbi.nlm.nih.gov/pmc/articles/PMC6937442/)
    • This suggests that a significant body of literature was not captured by the author's election to use the term "metastasis" as a mandatory inclusion in their search
  • Inappropriate keywords relative to the field
    • Please consider that a large amount of literature was missed because they did not index their literature with the most relevant terms
    • Please carefully consider your keywords to avoid the same

Direct suggestions and/or errors

  • Abstract -
    •  prognosticator is not formally a word used for this process, please consider changing to biomarker
      • This term is used in the main text as well and should be changed to fit with a formally accepted term
    • Radiomics is more than just the identification of imaging features, it's also their analysis and implementation  (https://pubs.rsna.org/doi/10.1148/radiol.2015151169)
    • Your data (line 186-188) suggested that an increase in homogeneity or skewness after chemotherapy correlated with a radiological tumour response, not and as noted in line 25
    • Grammar in the sentence spanning line 21-22
    • State the total number of records searched & the databases used
    • Line 28 implies you are looking at any role for radiomics (ex. Auto-segmentation; distinguishing from other mets)
      • However this is not your question in PROSPERO
      • By stating, I have interpreted your question is broader and when I reviewed your methods I found a substantial amount of data not included - there is a body of literature which seeks to separate HCC from other liver masses, which would technically include liver mets (as reference above in Limitations)
  • Background
    • Reference 6 does not add to your point in line 46
    • Lines 45-48, are a tad over-zealous - you're talking about a metastatic population, so thus adding in a liver met does not always worsen prognosis (ex. Testicular cancer; colorectal patients with peritoneal deposits); similarly, "Whenever possible" is too broad, please simply adjust your language to ensure your statements remain factual
    • Line 62 - radiomic features are biomarkers, this sentence should be amended to fit the intent of the statement
    • You have written this with a clinician's perspective (very valuable), but you have to provide adequate context of radiomic terms in the background
      • Please invest fewer words in the clinical merits of a biomarker - clinician's will already buy-in
      • Please describe key radiomics terms you use later (ex. Feature, texture, V/RoI)
  • Common terms
    • ROI is used far more commonly than VOI - this should be revised
  • Results
    • Unclear how you separated into the four groups in lines 98-99
      • For example, does prognosis mean overall survival? Any survival?
        • Re: grammar - predictive of prognosis is unclear, do you mean prognostic or predictive for survival
        • How does evaluation of therapy response differ from correlation with pathologic data
      • If you clarify this in your methods, please give us a "refer to section X.Y"
      • Were these distinctions set a priori or after reviewing your returns?
    • TRIPOD checklist - line 118, please reference this for your readers
    • Table 1
      • Please organize the presentation of radiomic features into a consistent format
    • Lines 136-145 - the values referred to should be succinctly summarized including a measure of an error
    • Tables require citations for the included studies
    • You state that Radiomic features out performed RECIST
  • Discussion
    • The beginning of the Discussion is more appropriate for the Introduction
      • Clinicians reading this may require Figure 3 to conceptualize everything you are presenting in Section 2.0
      • After a one-to-two sentence summary of your methods, Line 307 would be a better place to start your discussion
        • The material prior could be provided in a condensed introduction, as your word count there is ideal though much of the language is superfluous
    • For future directions, it would be worthwhile to point out:
      • how CNNs have not yet been used in this area - other MRI based radiomics platforms (ex. GBM and prostate) have been able to suggest significant gains in auto-segmentation & feature generation by implementing these features
      • How stability of MRI features is an uncertainty that must be addressed
      • The need for big data and the subsequent requirements in collaboration to resolve this
    • While the limitations in the evaluated studies were described, the limitations of this study's methodologies were not & there are numerous in any systematic review
      • Given the apparent choice to not observe a systematic format, these should be explicitly stated
  • Methods
    • Adequate description, though it is not clear that PRISMA guidelines were actually observed with the absence of supplemental materials being provided
    • The absence of a librarian in your methods has left a relevant hedge missing in your literature search
  • Abbreviations and formatting
    • When the paper was reformatted so that the Results preceded the methods, references and abbreviations were not adjusted
      • This is confusing, as a reader I am encountering terms and abbreviations I am not familiar with - this must be addressed

Round 2

Reviewer 3 Report

The major revisions are appropriate and I do not contest any of the author's rationale. I support your approach and your statement of observing PRISMA guidelines, with your now introduced caveats.

I have four minor revisions:

(1) When you first mention the RQS, please add the citation to it there for the reader's convenience.

(2) ROI can refer to 2D or 3D regions of interest - I would note that you do not have a citation to support your definitions. My initial comment was prompting you to simply use the more generic ROI rather than interchanging terms between ROI and VOI. The term VOI is not used nearly as commonly in the literature. At the very least, if you choose to use the term VOI, then please correct the error when you state that an ROI is 2D only OR provide a reputable reference to support your claim.

Whichever your decision, please ensure the terms remain consistent in your graphical abstract, please change VOI to ROI. 

(3) I do not believe you can use the term "better" in line 32/33 of the abstract based on the facts presented - thank you for presenting the point estimates and confidence intervals to facilitate this evaluation. I feel that the simplest resolution is to simply change the wording in the abstract to something gentler and more representative of the existing data.

Rationale to (3)

Your conclusion (abstract line 32-33) that radiomics models exceed the ability of RECIST is not supported by what you report in your results nor the single paper you explicitly point out in your results (Dohan et al.).  Dohan et al suggest that there is signal for this truth, but their comparisons do not show a significant difference (nor a robust enough trend) for one to reliably state they are "better" - their a priori endpoint was not met.  Dohan et al concluded that a combined radiomics model performed similarly to RECIST, not inferior to it as per the primary question.  Dohan et al's secondary endpoint discussion does suggest areas where there radiomics model seemed to be better than RECIST, but without powering their analysis for multiple testing, their restriction of their conclusions to their primary endpoint is a more robust extraction of their data.

If there were other results that support this "better" conclusion, it is not being communicated clearly in the Results nor Discussion. I fully agree with your communication of the potential of radiomics as a modality. From your paper, I only read that this one secondary endpoint is being used to to state that radiomics based models were better at predicting chemo but even this is not stated clearly.

(4) In regards to (2) and (3), please ensure the graphical abstract reflects any changes.
